# Quality assessment of clinical practice guidelines in Kenya using the AGREE II tool: a methodological review

Caleb Kimutai Sagam ⬡ ,[1] Lisa M Were,[1] Jenifer A Otieno ⬡ ,[1] Mercy N Mulaku,[1,2,3] Simon Kariuki,[1] Eleanor Ochodo ⬡ [1,2]

[1]Centre for Global Health Research, Kenya Medical Research Institute, Kisumu, Kenya
[2]Centre for Evidence-Based Health Care, Department of Global Health, Stellenbosch University, Cape town, South Africa
[3]Department of Pharmacology, Clinical Pharmacy, and Pharmacy Practice, Faculty of Health Sciences, University of Nairobi, Nairobi, Kenya

**Correspondence to**
Caleb Kimutai Sagam;
sagamc7@gmail.com

## ABSTRACT

**Objective** To assess the quality of available and accessible national Clinical Practice Guidelines (CPGs) in Kenya using the Appraisal of Guidelines for Research and Evaluation II (AGREE II) tool.

**Methods** We searched the websites of the Kenyan Ministry of Health, professional associations and contacted experts in relevant organisations. Our scope was guidelines on maternal, neonatal, nutritional disorders, injuries, communicable and non-communicable diseases in Kenya published in the last 5 years until 30 June 2022. Study selection and data extraction were done by three independent reviewers with disagreements resolved via discussion or with a senior reviewer. We conducted a quality assessment using the online English version of AGREE II tool across six domains. Descriptive statistics were analysed using Stata software V.17. The primary outcome was the methodological quality of the included CPGs assessed by the AGREE II tool score.

**Results** We retrieved 95 CPGs and included 24 in the analysis after screening for eligibility. The CPGs scored best in clarity of presentation and least in the rigour of development. In descending order, the appraisal scores (mean and CI) per domain were as follows: Clarity of presentation 82.96% (95% CI 78.35% to 87.57%) with all guidelines scoring above 50%. Scope and purpose 61.75% (95% CI 54.19% to 69.31%) with seven guidelines scoring less than 50%. Stakeholder involvement 45.25% (95% CI 40.01% to 50.49%) with 16 CPGs scoring less than 50%. Applicability domain 19.88% (95% CI 13.32% to 26.43%) with only one CPG scoring above 50%. Editorial independence 6.92% (95% CI 3.47% to 10.37%) with no CPG scoring above 50% and rigour of development 3% (95% CI 0.61% to 5.39%) with no CPG scoring at least 50%.

**Conclusion** Our findings suggest that the quality of CPGs in Kenya is limited mainly by the rigour of development, editorial independence, applicability and stakeholder involvement. Training initiatives on evidence-based methodology among guideline developers are needed to improve the overall quality of CPGs for better patient care.

## STRENGTHS AND LIMITATIONS OF THIS STUDY

⇒ We used the Appraisal of Guidelines for Research and Evaluation II (AGREE II) instrument, an internationally used and validated tool for the quality of Clinical Practice Guidelines (CPGs).
⇒ Three independent reviewers rigorously appraised the CPGs and the final score was reached through a consensus.
⇒ Poor reporting in identified CPGs limited our assessment with the AGREE II tool.
⇒ Due to absence of a guideline clearing house in Kenya, searches were limited to available guideline repositories, websites and experts.

According to the USA Institute of Medicine, CPGs are systematically developed recommendations that help healthcare practitioners make appropriate healthcare decisions to optimise patient care, informed by a systematic review of the evidence, assessment of benefits, harms and alternative care options.[2] Good quality evidence-based guidelines developed or adapted robustly and transparently are essential in ensuring good quality healthcare.[3]

Kenya, a low-middle income country with a life expectancy of about 66.7 years, is faced with health challenges, including a high burden of communicable diseases, an increasing burden of non-communicable diseases (NCDs), a high maternal and child mortality and injuries.[4] Communicable diseases have remained the leading cause of morbidity and mortality.[5] In 2020, mortality due to communicable diseases in the general population was 50.1%, a decline compared with 58.6% in 2016.[6] Mortality due to NCDs is on the rise, with 42.8% of deaths being attributed to NCDs in 2020, compared with 37% in 2016, while mortality attributed to injuries was 7.1% in 2020.[4] Four major NCDs, including cancer, diabetes, chronic respiratory and cardiovascular diseases, account for 33% of all the NCD deaths in Kenya, with

## INTRODUCTION

Clinical Practice Guidelines (CPGs) aid healthcare professionals in decision-making and provide a tool to implement recommended health policies at the point of care.[1]

other diseases such as mental health conditions on the increase.[7] Mortality is mainly high among infants and young children and those above the age of 65 years, a trend similar in both men and women.[6] According to the High-Quality Technical Assistance for Results organisation 2018 annual report, maternal and neonatal mortalities in Kenya account for 360/100 000 and 22/1000 live births, respectively.[8]

A skilled and well-guided health workforce is needed to tackle the outlined disease burden in Kenya. The WHO defines core healthcare workers as medical doctors, nurses and clinical officers/doctors assistants and recommends a minimum core health worker density threshold of 23 health workers per 10 000 population for effective service delivery.[9 10] As of 2020, Kenya's core health worker density was 16.6 health workers per 10 000, with 1.7 doctors per 10 000 population and 12.1 nurses per 10 000 population.[6] With a shortage of health workers amidst a high disease burden, well-developed, implemented and accessible CPGs are essential in guiding quality healthcare practices and improving the quality of life.

Previous studies have shown that CPGs improve health outcomes.[11] In a study by Flarity *et al*, CPGs were found to reduce hospital length of stay.[12] According to Ruseckaite *et al*, the use of CPGs in Australia led to an improvement in the nutritional status of children with cystic fibrosis.[13] Implementation strategies of the CPGs have also been shown to improve health outcomes.[3] An assessment of an international cohort of CPGs for glycaemic control in patients with type 2 diabetes mellitus highlighted the variation in the quality of CPGs and their effect on the quality of care.[14] To our knowledge, this is the first evaluation of the methodological quality of CPGs in Kenya. We aimed to assess the methodological quality of available and accessible national CPGs in Kenya using the Appraisal of Guidelines for Research and Evaluation II (AGREE II) tool[15] (online supplemental file 1). This review will inform the future development of high-quality evidence-based guidelines and recommend areas of strengthening the current CPGs in Kenya and beyond.

## METHODS
### Eligibility criteria
Data on mortality and morbidity in Kenya are sparse and the available data are often not linked together to produce a comprehensive ranking of the disease burden. We, therefore, included available and accessible Kenyan CPGs on diseases or conditions under the broad health categories identified by the global burden of diseases, injuries and risk factors study 2019.[16] The broad health categories included communicable diseases, non-communicable diseases, maternal disorders, neonatal disorders, nutritional disorders and injuries. We included the latest CPGs versions for screening, diagnosis and treatment/management used in direct patient care for adults and children. We included CPGs published in English in the last 5 years

until 30 June 2022. It is estimated that CPGs get outdated 5 years after their development.[1]

We excluded guideline reports (manuscripts and any other published papers), summaries, press statements or interim guidance documents because they are typically temporary and intended for provisional or emergency situations. Due to their urgent nature a rigorous evidence-based methodology may not be used hence they could have incomplete evidence or guidance with methods not reported fully.[17] We also excluded health system guidelines because they are not used directly in patient care.

### Identification of clinical guidelines
Kenya does not have a guideline clearing house to enable identification of all published and current health guidelines. CPGs are also rarely published in the scientific literature. We thus searched the websites of the Kenyan Ministry of Health, Health Associations and contacted relevant organisations' personnel for additional guidelines. We also searched the Kenya e-repository (for Kenyan government documents). We applied broad searches in Google, using the following search terms 'clinical practice guidelines', 'health guidelines' and 'Kenya'. We searched for guidelines in the websites of the ministry of health and various health associations by clicking the guidelines, standards or policy links. Guidelines were searched with the cut-off date of 30 July 2022. Only guidelines published within the last 5 years from this cut-off date were included. A detailed list of websites, databases, type of search and full search strategy used in Google can be found in online supplemental file 2.

### Patient and public involvement
The study design did not include patient and public involvement.

### Screening and selection
We conducted CPG's screening and selection using an Excel spreadsheet, where we arranged all retrieved guidelines according to diseases and the latest guideline for each disease was picked for appraisal. CKS, JAO and LMW screened the title and publication year of CPG documents for eligibility. We retrieved CPGs that meet the eligibility criteria for quality assessment. We resolved discrepancies on inclusion through consensus or discussion with EO. Selection process details are provided in the Preferred Reporting Items for Systematic Reviews and Meta-Analysis 2020 flow chart below.[18] The protocol for this review was registered in Open Science Framework (OSF).[19]

### Data extraction
CKS, JAO and LMW independently extracted data using a Google form into an Excel worksheet (online supplemental file 3). We extracted the general information of the included guidelines, including title, year of publication, author, disease and scope of the guideline. We resolved discrepancies through consensus or discussion with EO.

## Quality assessment

CKS, JAO and LMW independently assessed the quality of CPGs using the AGREE II tool, a validated and widely used tool, recommended by WHO for assessing the quality, transparency and rigour of CPGs.[15] An international group of guideline developers and researchers first developed AGREE II in 2003. It was published in 2009, revised in 2017 and is currently used globally in appraising CPGs.[20] It contains six domains and 23 items.[15] The six domains evaluated include scope and purpose, stakeholder involvement, the rigour of development, (evidence-based methodology), clarity of presentation, applicability and editorial independence (conflict of interest).

The reviewers conducted the appraisal and scored each item against a 7-point Likert scale on the online version of AGREE II. We scored 1 if none of the criteria for an item was met or the item was reported very poorly. We scored 7 when the item met all the criteria and was well-reported.

The primary outcome of this study was the methodological quality of the CPGs assessed by AGREE II score. The quality scores were determined by calculating the scaled domain percentages for each CPG as outlined in the AGREE II tool.[15] We have reported each item's average appraiser score and overall assessment scores. We also calculated the average scores for the 23 items and the overall assessment score for each CPG using Stata software V.17 (Stata Corp, College Station, Texas, USA). We present our results in tables as means, CIs, SDs and graphs. The overall assessment score is not an average of individual domain scores but an independent domain.

## RESULTS

### Search results

We retrieved 95 CPGs from the Ministry of Health Kenya, different divisions and health associations' websites. After the removal of duplicates, we identified 61 guidelines. During full-text screening, we excluded 37 guidelines that did not meet the inclusion criteria. We finally included 24 CPGs (figure 1). A list of excluded CPGs can be found in online supplemental file 4.

### General characteristics of included guidelines

The description of the general characteristics of the assessed guidelines is illustrated in (table 1). Most of the guidelines were integrated (these are guidelines containing more than one purpose, that is, Diagnosis and screening and treatment or management) (n=19 CPGs). Six CPGs were published in 2017, six in 2018, two in 2019, five in 2020, four in 2021 and one in 2022. The Ministry of Health Kenya authored the majority of the guidelines.

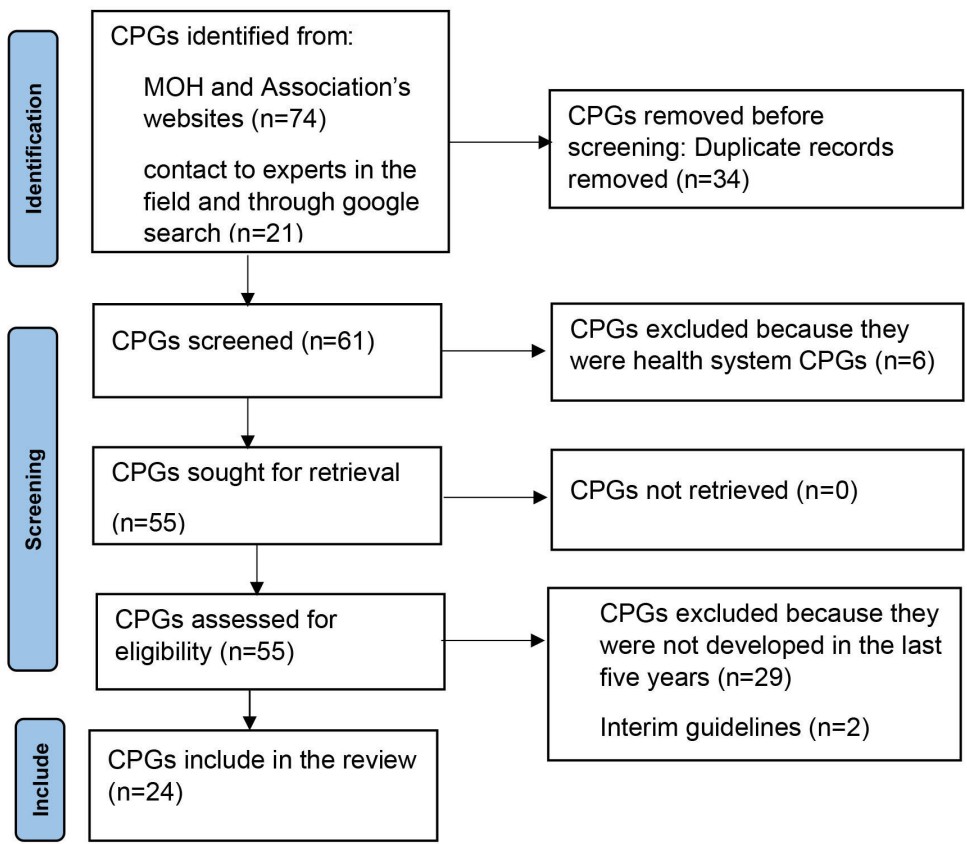

**Figure 1** Preferred Reporting Items for Systematic Reviews and Meta-Analysis 2020 flow diagram showing CPGs inclusion process. CPGs, Clinical Practice Guidelines; MOH, Ministry of Health.

**Table 1** Characteristics of included clinical practice guidelines

| Disease domain | Title of the guideline | Author | Year of publication | Scope |
|---|---|---|---|---|
| COVID-19 | Guidelines on case management of COVID-19 in Kenya[45] | MoH | 2021 | Integrated |
| | Guidelines to be used by occupational therapists in rehabilitation of patients during COVID-19 pandemic[19] | MoH | 2020 | Management |
| HIV | Guidelines on use of antiretroviral drugs for treating and preventing HIV infection in Kenya[46] | MoH and NASCOP | 2018 | Integrated |
| | National guidance on tetanus prevention in voluntary medical male circumcision settings in Kenya[47] | MoH | 2017 | Prevention |
| TB | National guidelines on management of tuberculosis in children[48] | MoH | 2017 | Integrated |
| | Integrated guideline for tuberculosis, leprosy and lung disease[49] | MoH | 2021 | Integrated |
| | Field guide on systematic screening of active TB in Kenya[50] | MoH | 2017 | Screening |
| Malaria | Guidelines for the diagnosis, treatment & prevention of malaria in Kenya[51] | MoH | 2020 | Integrated |
| CVD | Kenya national guidelines for cardiovascular diseases management[52] | MoH | 2018 | Integrated |
| Eye | National guidelines for the management of glaucoma[53] | MoH | 2020 | Integrated |
| | Guidelines for the screening and management of retinopathy of prematurity in Kenya[54] | MoH | 2018 | Integrated |
| Diabetes | National clinical guidelines for management of diabetes mellitus[55] | MoH | 2018 | Integrated |
| | Guidelines for screening and management of diabetic retinopathy[56] | MoH | 2017 | Integrated |
| Cancer | Kenya national cancer treatment protocols[57] | MoH | 2019 | Integrated |
| | Retinoblastoma best practice guidelines[58] | MoH | 2019 | Integrated |
| | National cancer screening guidelines[59] | MoH | 2018 | Screening |
| Renal disease | Guidelines for the management of emergencies in dialysis[60] | KRA | 2017 | Management |
| Mental health | National tele-mental health guidelines[61] | MoH | 2021 | Integrated |
| Respiratory diseases | Kenya asthma management guideline[62] | MoH | 2021 | Integrated |
| Maternal health | Guidelines for postnatal care[63] | MoH | 2017 | Integrated |
| Paediatrics | Guidelines on the management of paediatric patients during COVID-19 pandemic[64] | MoH | 2020 | Integrated |
| | Basic paediatric protocols[65] | MoH | 2022 | Integrated |
| Sickle cell | National guidelines for control and management of sickle cell disease in Kenya[66] | MoH | 2020 | Integrated |
| Nutrition | National guidelines for healthy diets and physical activity[67] | MoH | 2017 | Prevention |

CVD, cardiovascular diseases; KRA, Kenya Renal Association; MoH, Ministry of Health; NASCOP, National AIDS and STIs Control Programme; TB, tuberculosis.

## Findings on AGREE II guideline quality score per domain

In reference to table 2, figure 2 and online supplemental file 5, we present the findings from the highest-scoring domain to the least-scoring domain.

## Clarity of presentation

The mean score was 82.96% (95% CI 78.35% to 87.57%) with all guidelines scoring above 50%. The overall mean of this domain was the highest among all domains. Recommendations were specific and unambiguous. Key recommendations were easily identifiable and different options for managing the health condition were stated

**Table 2** Domain scores for each CPG

| CPG name | Domain percentages scores | | | | | | Average (%) |
|---|---|---|---|---|---|---|---|
| | 1 | 2 | 3 | 4 | 5 | 6 | |
| Guidelines for screening and management of diabetic retinopathy[56] | 96 | 72 | 26 | 81 | 42 | 28 | 67 |
| National guidelines for the management of glaucoma[53] | 98 | 63 | 11 | 96 | 29 | 0 | 61 |
| Integrated guideline for tuberculosis, leprosy and lung disease[49] | 67 | 56 | 1 | 93 | 42 | 8 | 56 |
| Guidelines for the screening and management of retinopathy of prematurity in Kenya[54] | 81 | 63 | 5 | 91 | 31 | 8 | 56 |
| National guidelines for healthy diets and physical activity[67] | 81 | 44 | 0 | 94 | 53 | 3 | 56 |
| Field guide on systematic screening of active TB in Kenya[50] | 56 | 30 | 1 | 89 | 49 | 0 | 50 |
| National cancer screening guidelines[59] | 70 | 54 | 2 | 94 | 7 | 11 | 50 |
| Kenya asthma management guideline[62] | 78 | 52 | 1 | 87 | 38 | 3 | 50 |
| National guidelines for control and management of sickle cell disease in Kenya[66] | 70 | 54 | 1 | 85 | 13 | 0 | 50 |
| National guidance on tetanus prevention in voluntary medical male circumcision settings in Kenya[47] | 44 | 30 | 8 | 72 | 29 | 17 | 44 |
| National guidelines on management of tuberculosis in children[48] | 69 | 52 | 0 | 76 | 19 | 0 | 44 |
| Guidelines for the diagnosis, treatment & prevention of malaria in Kenya[51] | 70 | 44 | 1 | 91 | 14 | 14 | 44 |
| Kenya national cancer treatment protocols[57] | 44 | 50 | 0 | 94 | 11 | 6 | 44 |
| Retinoblastoma best practice guidelines[58] | 54 | 41 | 4 | 85 | 8 | 3 | 44 |
| Guidelines on the management of pediatric patients during COVID-19* pandemic[64] | 57 | 28 | 6 | 80 | 0 | 0 | 44 |
| National clinical guidelines for management of diabetes mellitus[55] | 63 | 46 | 0 | 83 | 11 | 25 | 44 |
| Basic pediatric protocols[65] | 50 | 41 | 1 | 78 | 18 | 6 | 44 |
| Guidelines to be used by occupational therapists in rehabilitation of patients during COVID-19* pandemic[19] | 65 | 52 | 2 | 80 | 7 | 0 | 39 |
| Kenya national guidelines for cardiovascular diseases management[52] | 37 | 37 | 0 | 93 | 14 | 6 | 39 |
| Guidelines for postnatal care[63] | 65 | 48 | 0 | 78 | 17 | 3 | 39 |
| Guidelines on case management of COVID-19* in Kenya[45] | 56 | 30 | 1 | 81 | 4 | 0 | 33 |
| Guidelines on the use of antiretroviral drugs for treating and preventing HIV infection in Kenya[46] | 48 | 44 | 0 | 80 | 7 | 19 | 33 |
| National tele-mental health guidelines[61] | 35 | 31 | 0 | 54 | 14 | 6 | 33 |
| Guidelines for management of emergencies in dialysis[60] | 28 | 24 | 1 | 56 | 0 | 0 | 28 |

COVID-19, coronavirus disease 2019; CPG, Clinical Practice Guideline; TB, tuberculosis.

in varying ways. Items 15 and 16 scored above 6 and 17 scored above 5 (table 3).

## Scope and purpose

The mean score in this domain was 61.75% (95% CI 54.19% to 69.31%). Most guidelines scored well in this domain with only seven scoring less than 50%. Under domain 1, items 1, 2 and 3 scored above 4 out of the possible 7. This is because the health questions, the guideline objective and the population to whom the guideline was applied were described in most guidelines. However, adequate information was not reported on the expected benefit or outcome and healthcare setting. The description of health intents was not well written and clear in some guidelines.

## Stakeholder involvement

The mean score in this domain was 45.25% (95% CI 40.01% to 50.49%). This domain score was lower than domain 1 because most guidelines did not involve evidence-based experts and patients in the development process. Sixteen CPGs scored less than 50%. In domain 2, items 4 and 6 scored above 4 out of possible 7 (table 3). Item 5 had a score of 1, this is because the views and preferences of the target population were only sought in 2 out of the 24 CPGs. Disciplines and a description of the member's role in the guideline development group were missing in most guidelines.

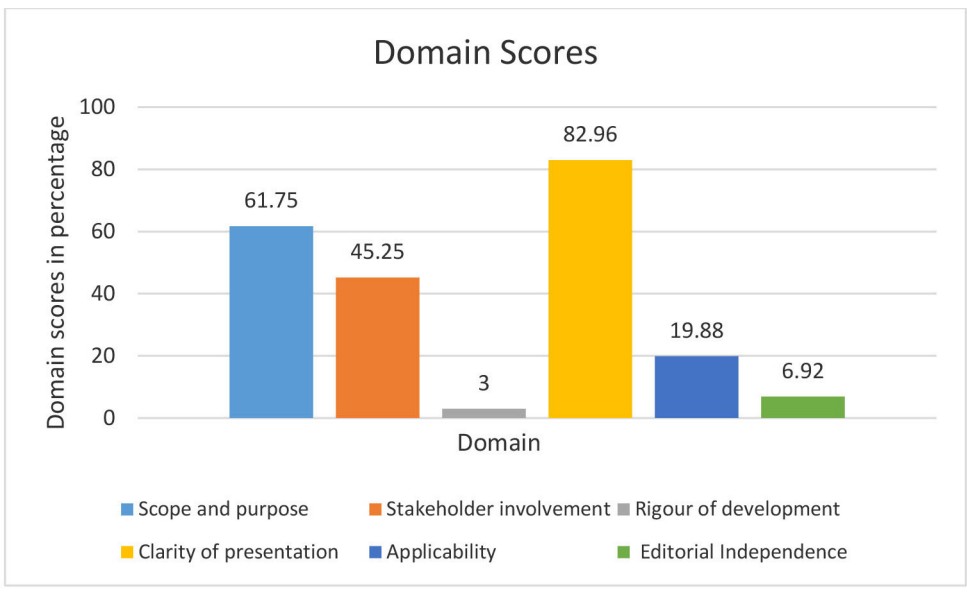

**Figure 2** Mean domain scores of clinical practice guidelines.

**Table 3** Quality of CPGs per item of the AGREE II tool

| Variable | Mean | SD | (95% confidence interval) |
|---|---|---|---|
| Item 1 | 5.03 | 1.43 | (4.42 to 5.63) |
| Item 2 | 4.06 | 1.57 | (3.39 to 4.72) |
| Item 3 | 4.54 | 1.25 | (4.01 to 5.07) |
| Item 4 | 4.26 | 1.35 | (3.69 to 4.84) |
| Item 5 | 1.32 | 0.68 | (1.03 to 1.61) |
| Item 6 | 5.66 | 1.49 | (4.93 to 6.18) |
| Item 7 | 1.13 | 0.35 | (0.98 to 1.27) |
| Item 8 | 1.07 | 0.22 | (0.98 to 1.16) |
| Item 9 | 1.01 | 0.07 | (0.99 to 1.04) |
| Item 10 | 1.11 | 0.29 | (0.99 to 1.23) |
| Item 11 | 1.08 | 0.28 | (0.96 to 1.20) |
| Item 12 | 1.24 | 0.71 | (0.94 to 1.54) |
| Item 13 | 1.14 | 0.37 | (0.98 to 1.29) |
| Item 14 | 1.68 | 1.35 | (1.11 to 2.25) |
| Item 15 | 6.10 | 0.79 | (5.76 to 6.43) |
| Item 16 | 6.32 | 0.75 | (6.00 to 6.63) |
| Item 17 | 5.61 | 0.95 | (5.21 to 6.01) |
| Item 18 | 1.61 | 0.93 | (1.22 to 2.00) |
| Item 19 | 2.94 | 0.98 | (2.53 to 3.36) |
| Item 20 | 1.49 | 0.85 | (1.13 to 1.85) |
| Item 21 | 2.71 | 1.95 | (1.89 to 3.53) |
| Item 22 | 1.82 | 0.98 | (1.40 to 2.23) |
| Item 23 | 1.00 | 0.00 | (1.00 to 1.00) |
| Overall assessment | 3.74 | 0.56 | (3.50 to 3.97) |

AGREE II, Appraisal of Guidelines for Research and Evaluation II; CPGs, Clinical Practice Guidelines.

### Applicability

The applicability domain mean score was 19.88% (95% CI 13.32% to 26.43%) with only one CPG scoring above 50%. Most of the CPGs did not describe the facilitators and barriers of its application or provide advice on how tools can be put into practice. They also did not present monitoring and evaluation criteria and potential resource implications of the recommendations. Under domain 5, all the items scored below 3 out of the possible 7, especially in items 18 and 20 (table 3).

### Editorial independence

The mean score was 6.92% (95% CI 3.47% to 10.37%) with no CPG scoring above 50%. The scores in this domain were the second lowest after the rigour of development. Items in this domain scored less than 2, this is because information on funding was either not stated or not clear. Conflict of interest was poorly reported, this contributed to the low score in domain six because information on the declaration of competing interest by the guideline development group was not provided. Though a third of the guidelines mentioned the funding bodies, the role of the funding bodies in the guideline development process was not provided.

### Rigour of development

This domain has the highest number of items and plays a key role in developing CPGs. The mean was 3% (95% CI 0.61% to 5.39%) with no CPG scoring at least 50%. The overall domain score was poor with the highest score of 26%. A total of eight CPGs scored 0% in this domain which points out serious shortcomings.

Under domain 3, all the items from items 7 to 14 scored less than 2 out of the possible 7 (table 3). Systematic methods were not used to search for evidence, the criteria for selecting the evidence were not described and the strengths and limitations of the evidence were not

described. In addition, methods for formulating recommendations were not clear, there was no information on whether health benefits, side effects, harms and risks were considered in formulating recommendations. An explicit link between recommendation and supporting evidence was also not presented and there was no information to show that the guideline was externally reviewed before its publication. All these factors led to a score of less than 2 for each item.

In this study, the scores of the 24 CPGs on the six domains, the only two domains scored above 50% is scope and purpose and clarity of presentation. These guidelines scored low in editorial independence, applicability and rigour of development.

## DISCUSSION

We assessed the quality of 24 CPGs in Kenya published in the last 5 years until 30 June 2022 using the AGREE II tool across six domains: clarity of presentation, scope and purpose, stakeholder involvement, applicability, editorial independence and rigour of development. The CPGs scored best in clarity of presentation (82.96% (95% CI 78.35% to 87.57%)) and scored least in the rigour of development (3% (95% CI 0.61% to 5.39%)).

Similar to our results, a study done in China reported challenges in the rigour of development.[21] The quality of CPGs is heavily affected by the use of evidence-based methods.[22] In a systematic review of the effect of evidence-based CPGs on the quality of care, evidence-based clinical guidelines improved the quality of care.[3] It is recommended that guideline recommendations be made based on the best available evidence using an evidence-to-decision framework (EtD).[23] EtD frameworks (such as the Grading of Recommendations Assessment, Development and Evaluation (GRADE EtD) and WHO INTEGRATE (EtD framework rooted in WHO norms and values that is, in principle, suitable for individual-level, population-level and system-level health interventions that may or may not be characterised by complexity) provide a systematic and explicit process linking the best available evidence to formulating the recommendations.[24–26]

According to a study on publishing CPGs, external review plays a critical role in enhancing scientific validity, clarity and feasibility of the guidelines before formal publication.[27] External review is a critical step in enhancing accountability during guideline development. It provides an opportunity for additional inputs and a critical review of the guideline by a specialist before its publication. Our findings show that over two-thirds of the guidelines did not report on the procedure and frequency of updating the guideline. The body of knowledge is constantly changing, therefore there is a need to review CPGs. Updating CPGs is very important as it enhances the validity of the recommendations.[28] According to a systematic review of guidance for updating CPGs 2–3 years is the recommended time frame between publishing a guideline and commencing the updating process.[29]

Similar to our results, a study done in China found shortcomings in editorial independence.[21] This can be attributed to conflict-of-interest disclosure policies and whether they have been developed or challenges that come with implementation. Transparency is a key aspect in the development of CPGs.[27] According to a study on reporting financial conflicts of interest in CPGs, conflict of interest is a threat because adequate literature is documented on the influence of uncontrolled conflict of interest on recommendations.[30] The influence of external activities such as grant funding and ownership of commercial entities have created a bias and affected the process of making recommendations.[31] Efforts to balance guideline development groups (GDGs) have improved however included experts in the industry involved in commercial activities which makes it difficult to rule out a conflict of interest.[27] Reporting is also an integral part of ensuring transparency, because when a conflict of interest was done but not reported it brings confusion and it will always be assumed that it was not done which may not be the reality.[32]

Poor applicability threatens the usability of the guidelines at the facility level. The applicability domain goes beyond the methodological quality and covers resource implication, which is key to successfully implementing each guideline.[33] Similar to our study on appraisal of clinical guidelines for recurrent urinary tract infections also showed low scores in this domain.[29] Monitoring and evaluation plan ensures that the guidelines serve their intended purpose. In many cases the guideline developers may not be responsible for the direct implementation of the guideline therefore it is important to stipulate how the monitoring and auditing will be done to ensure adequate implementation of the guideline.

GDG composition is one of the most important aspects of guideline development.[26] A GDG necessitates the expertise of healthcare professionals, patient input on their needs and preferences and methodologists and librarians skilled in gathering, summarising and interpreting evidence.[34] Structured education in this methodological area will be required, given the increasing demands and expertise required to manage complex guideline development projects.[35] The focus of the guidelines should influence the number of group members and the balance of disciplines. When deciding on the group's composition, all potential stakeholders should be identified, including healthcare professionals who are directly involved in the clinical management of patients in various healthcare settings (eg, primary and secondary care), policymakers who may need to make resource usage decisions and patients.[35] The decision must then be made about which categories of participants to include in the guideline group. Guidelines developers must frequently balance the desire for broad representation with the need for a cohesive working group. Small groups may lack experience among their members, while larger groups may lack cohesiveness and be difficult to lead.[23]

## Limitations of the study

Our study only included available guidelines published in the last 5 years on the broad health categories; communicable diseases, non-communicable diseases, maternal disorders, neonatal disorders, nutritional disorders and injuries. This may have excluded other key guidelines. There is also no clear way of differentiating appraisal scores in cases where there was poor reporting and no reporting, this is because both are rated 1 according to AGREE II. The tool also relies on the subjective judgements of the assessors, which may vary depending on their expertise and expectations.

Accessibility of some guidelines was also a challenge due to lack of a common repository for CPGs or guideline clearing house in Kenya. Furthermore, Kenyan guidelines are not indexed in the electronic databases that are commonly used, such as PubMed or Cochrane Library, therefore we limited our search to Google, websites and contacting experts. With the limited sources to retrieve CPGs we did not involve an information specialist but employed broad searches. With this our search may have been limited but we believe we identified a representative sample to judge the quality of existing CPGs.

## Implication for practice

Given our findings, we recommend that awareness of the Reporting Items for Practice Guidelines in HealThcare (RIGHT) statement among guideline developers be improved.[36] This could be done through conferences, webinars and training workshops by evidence-based groups in Africa (such as Guidelines International Network,[37 38] Cochrane Africa,[39 40] Africa Evidence Network,[41] Joanna Briggs Institute[42 43] and GRADE networks in Africa[44]). Increased awareness will enable better and more transparent reporting of CPGs. More training initiatives among GDG members on aspects of AGREE II and RIGHT guidelines and the involvement of a guideline methodologist will improve the rigour of development, reporting and overall methodological quality of CPGs in Kenya.

## Implications for research

A qualitative study to explore guideline developers' knowledge and perspectives on guideline development methods would be useful in knowing the barriers and enablers to guideline development in Kenya. Qualitative research among the users of the guidelines (healthcare workers) would also shed light on barriers to the impact of the guidelines on health practices and outcomes. Impact evaluations on the effect of low-quality or high-quality CPGs on healthcare in Kenya and other low-middle income country settings are also needed.

## Conclusion

The quality of CPGs in Kenya is limited mainly by the rigour of development, editorial independence, applicability and stakeholder involvement. Training initiatives on evidence-based methodology among guideline developers are needed to improve the overall quality of CPGs for better patient care.

**Contributors** Conceptualisation of the study: EO. Writing of the first draft: CKS, JAO and LMW. Critical review of the manuscript: JAO, LMW, MNM, SK and EO. Writing the final draft of the manuscript: CS. Approval of the final draft of the manuscript: CKS, JAO, LMW, MNM, SK and EO. EO is the guarantor for this article.

**Funding** This study is funded under the UK MRC African Research Leaders award number MR/T008768/1. The funding organisations had no role in the development of this study.

**Competing interests** None declared.

**Patient and public involvement** Patients and/or the public were not involved in the design, or conduct, or reporting, or dissemination plans of this research.

**Patient consent for publication** Not applicable.

**Ethics approval** We did not seek formal ethical approval since this is a methodological review based on publicly available Clinical Practice Guidelines.

**Provenance and peer review** Not commissioned; externally peer reviewed.

**Data availability statement** Data are available upon reasonable request.

**ORCID iDs**
Caleb Kimutai Sagam http://orcid.org/0000-0002-7316-3340
Jenifer A Otieno http://orcid.org/0000-0001-9521-624X
Eleanor Ochodo http://orcid.org/0000-0002-7951-3030

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
