## [Reviewer comments · BMJ Open]

ARTICLE DETAILS

TITLE (PROVISIONAL)	Quality Assessment of Clinical Practice Guidelines in Kenya Using the AGREE II Tool:A Methodological Review
AUTHORS	Sagam, Caleb; Were, Lisa M.; Otieno, Jenifer A.; Mulaku, Mercy N.; Kariuki, Simon; Ochodo, Eleanor

VERSION 1 – REVIEW

REVIEWER	Turner, Tari Monash University, School of Public Health and Preventive Medicine
REVIEW RETURNED	02-May-2023

GENERAL COMMENTS	This is an interesting, well conducted and well reported study. The paper would be improved by an explicit discussion of the limitations of the research – you might reflect on how effective your search approach was, for example. A few other minor issues for your consideration - In Eligibility criteria there is a statement “We excluded guideline reports (manuscripts)”. Do you mean that you excluded published journal papers (presumably because you thought they were too short to contain the relevant information)? I think this requires a strengthened rationale. Did you, for instance, use these papers to identify the full guideline documents and then source and include those? Similarly, why were interim guidelines excluded? Did you seek to identify the finalised versions of these guidelines?- The Google search appears quite limited. Was an information specialist involved in designing the search approach?- I am very glad you only present the individual domain scores, and not an overall score. That is appropriate. Thank you.
--

REVIEWER	Hatakeyama, Yosuke Toho University, School of Medicine
REVIEW RETURNED	02-May-2023

GENERAL COMMENTS	- This study described in detail the current state of the quality of CPGs in Kenya, and pointed out domains should be improved in the future.-However, the implications of this study could be presented in advance of the results, and the scope of implication is limited to Kenya (LMIC, at best). It was essential to clarify the relationship between the results of this study and the vast amount of quality evaluation on CPGs conducted in previous studies, and to reexamine the implications of this study.
---

VERSION 1 – AUTHOR RESPONSE

Reviewer: 1

4. The paper would be improved by an explicit discussion of the limitations of the research – you might reflect on how effective your search approach was, for example.

We agree. Thank you for the suggestion.

We have included the limitations of the study in the discussion under the subtitle limitation of the study.

5. In Eligibility criteria there is a statement “We excluded guideline reports (manuscripts)”. Do you mean that you excluded published journal papers (presumably because you thought they were too short to contain the relevant information)? I think this requires a strengthened rationale.

Thank you for the comment.

This study mainly focused on finalized clinical practice guidelines. We therefore excluded any other document that was not a finalised clinical guideline including published journal papers because they do not contain all the information of the guideline. During our search we were not able to identify any manuscript for the Kenyan clinical practice guidelines.

6. Did you, for instance, use these papers to identify the full guideline documents and then source and include those?

Thank you for the comment.

During our search we were not able to identify any manuscript for the Kenyan clinical practice guidelines.

7. Similarly, why were interim guidelines excluded? Did you seek to identify the finalised versions of these guidelines?

Thank you for the comment.

We excluded interim guidelines because interim guidelines are intended to provide a temporary framework for the implementation of the new policy. They are not meant to be exhaustive or definitive, and they may be revised or updated as needed. The interim guidelines have some limitations such as they do not cover all possible scenarios or situations that may arise in the course of the policy execution. They also do not reflect the feedback or input from all relevant stakeholders or experts.

We therefore did not want to include guidelines with already known limitations.

We identified finalized guidelines for all interim guidelines where applicable.

8. The Google search appears quite limited. Was an information specialist involved in designing the search approach?

Thank you for the question.

Because of limited sources to retrieve CPGs in Kenya, we did not involve an information specialist but employed broad searches. We have however highlighted this as one of the limitations of our study. We have revised supplementary file 2 and included a search strategy that was adopted in searching Google.

Reviewer: 2

9. However, the implications of this study could be presented in advance of the results, and the scope of implication is limited to Kenya (LMIC, at best). It was essential to clarify the relationship between the results of this study and the vast amount of quality evaluation on CPGs conducted in previous studies, and to reexamine the implications of this study. We agree. Thank you for the comment.

We have limited implications to Kenya and LMIC.

However, there are evidence-based groups in Africa that play a key role in knowledge dissemination in the entire Africa therefore they are key.

The relationship between the results of this study and the vast amount of quality evaluation on CPGs conducted in previous studies is captured in our discussion.